# MEMORIZATION-DILATION:
# MODELING NEURAL COLLAPSE UNDER LABEL NOISE

**Duc Anh Nguyen**[*]
LMU Munich

**Ron Levie**[*]
Technion
Israel Institute of Technology

**Julian Lienen**[*]
Paderborn University

**Gitta Kutyniok**
LMU Munich
University of Tromsø

**Eyke Hüllermeier**
LMU Munich

## ABSTRACT

The notion of neural collapse refers to several emergent phenomena that have been empirically observed across various canonical classification problems. During the terminal phase of training a deep neural network, the feature embedding of all examples of the same class tend to collapse to a single representation, and the features of different classes tend to separate as much as possible. Neural collapse is often studied through a simplified model, called the layer-peeled model, in which the network is assumed to have "infinite expressivity" and can map each data point to any arbitrary representation. In this work we study a more realistic variant of the layer-peeled model, which takes the positivity of the features into account. Furthermore, we extend this model to also incorporate the limited expressivity of the network. Empirical evidence suggests that the memorization of noisy data points leads to a degradation (dilation) of the neural collapse. Using a model of the memorization-dilation (M-D) phenomenon, we show one mechanism by which different losses lead to different performances of the trained network on noisy data. Our proofs reveal why label smoothing, a modification of cross-entropy empirically observed to produce a regularization effect, leads to improved generalization in classification tasks.

## 1   INTRODUCTION

The empirical success of deep neural networks has accelerated the introduction of new learning algorithms and triggered new applications, with a pace that makes it hard to keep up with profound theoretical foundations and insightful explanations. As one of the few yet particularly appealing theoretical characterizations of overparameterized models trained for canonical classification tasks, *Neural Collapse* (NC) provides a mathematically elegant formalization of learned feature representations Papyan et al. (2020).

To explain NC, consider the following setting. Suppose we are given a *balanced* dataset $\mathcal{D} = \left\{(\boldsymbol{x}_n^{(k)}, y_n)\right\}_{k \in [K], n \in [N]} \subset \mathcal{X} \times \mathcal{Y}$ in the instance space $\mathcal{X} = \mathbb{R}^d$ and label space $\mathcal{Y} = [N] := \{1, \ldots, N\}$, i.e. each class $n \in [N]$ has exactly $K$ samples $\boldsymbol{x}_n^{(1)}, \ldots, \boldsymbol{x}_n^{(K)}$. We consider network architectures commonly used in classification tasks that are composed of a *feature engineering* part $g \colon \mathcal{X} \to \mathbb{R}^M$ (which maps an input signal $\boldsymbol{x} \in \mathcal{X}$ to its feature representation $g(\boldsymbol{x}) \in \mathbb{R}^M$) and a *linear classifier* $\boldsymbol{W}(\cdot) + \boldsymbol{b}$ given by a weight matrix $\boldsymbol{W} \in \mathbb{R}^{N \times M}$ as well as a bias vector $\boldsymbol{b} \in \mathbb{R}^N$. Let $\boldsymbol{w}_n$ denote the row vector of $\boldsymbol{W}$ associated with class $n \in [N]$. During training, both classifier components are simultaneously optimized by minimizing the cross-entropy loss.

---

[*]These authors contributed equally to this work.

Denoting the *feature representations* $g(\boldsymbol{x}_n^{(k)})$ of the sample $\boldsymbol{x}_n^{(k)}$ by $\boldsymbol{h}_n^{(k)}$, the *class means* and the *global mean* of the features by

$$\boldsymbol{h}_n := \frac{1}{K} \sum_{i=1}^{K} \boldsymbol{h}_n^{(k)}, \qquad \boldsymbol{h} := \frac{1}{N} \sum_{n=1}^{N} \boldsymbol{h}_n,$$

NC consists of the following interconnected phenomena (where the limits take place as training progresses):

(NC1) **Variability collapse.** For each class $n \in [N]$, we have $\frac{1}{K} \sum_{k=1}^{K} \left\| \boldsymbol{h}_n^{(k)} - \boldsymbol{h}_n \right\|^2 \to 0$.

(NC2) **Convergence to simplex equiangular tight frame (ETF) structure.** For any $m, n \in [N]$ with $m \neq n$, we have

$$\|\boldsymbol{h}_n - \boldsymbol{h}\|_2 - \|\boldsymbol{h}_m - \boldsymbol{h}\|_2 \to 0, \text{ and}$$
$$\left\langle \frac{\boldsymbol{h}_n - \boldsymbol{h}}{\|\boldsymbol{h}_n - \boldsymbol{h}\|_2}, \frac{\boldsymbol{h}_m - \boldsymbol{h}}{\|\boldsymbol{h}_m - \boldsymbol{h}\|_2} \right\rangle \to -\frac{1}{N-1}.$$

(NC3) **Convergence to self-duality.** For any $n \in [N]$, it holds

$$\frac{\boldsymbol{h}_n - \boldsymbol{h}}{\|\boldsymbol{h}_n - \boldsymbol{h}\|_2} - \frac{\boldsymbol{w}_n}{\|\boldsymbol{w}_n\|_2} \to 0.$$

(NC4) **Simplification to nearest class center behavior.** For any feature representation $\boldsymbol{u} \in \mathbb{R}^M$, it holds

$$\arg\max_{n \in [N]} \langle \boldsymbol{w}_n, \boldsymbol{u} \rangle + b_n \to \arg\min_{n \in [N]} \|\boldsymbol{u} - \boldsymbol{h}_n\|_2.$$

In this paper, we consider a well known simplified model, in which the features $\boldsymbol{h}_n^{(k)}$ are not parameterized by the feature engineering network $g$ but are rather free variables. This model is often referred to as *layer-peeled* model or *unconstrained features* model, see e.g. Lu & Steinerberger (2020); Fang et al. (2021); Zhu et al. (2021). However, as opposed to those contributions, in which the features $\boldsymbol{h}_n^{(k)}$ can take any value in $\mathbb{R}^M$, we consider here the case $\boldsymbol{h}_n^{(k)} \geq 0$ (understood component-wise). This is motivated by the fact that features are typically the outcome of some non-negative activation function, like the Rectified Linear Unit (ReLU) or sigmoid. Moreover, by incorporating the limited expressivity of the network to the layer-peeled model, we propose a new model, called *memorization-dilation* (MD). Given such model assumptions, we formally prove advantageous effects of the so-called label smoothing (LS) technique Szegedy et al. (2015) (training with a modification of cross-entropy (CE) loss), in terms of generalization performance. This is further confirmed empirically.

## 2 RELATED WORK

Studying the nature of neural network optimization is challenging. In the past, a plethora of theoretical models has been proposed to do so Sun (2020). These range from analyzing simple linear Kunin et al. (2019); Zhu et al. (2020); Laurent & von Brecht (2018) to non-linear deep neural networks Saxe et al. (2014); Yun et al. (2018). As one prominent framework among others, Neural Tangent Kernels Jacot et al. (2018); Roberts et al. (2021), where neural networks are considered as linear models on top of randomized features, have been broadly leveraged for studying deep neural networks and their learning properties.

Many of the theoretical properties of deep neural networks in the regime of overparameterization are still unexplained. Nevertheless, certain peculiarities have emerged recently. Among those, so-called "benign overfitting" Bartlett et al. (2019); Li et al. (2021), where deep models are capable of perfectly fitting potentially noisy data by retaining accurate predictions, has recently attracted attention. Memorization has been identified as one significant factor contributing to this effect Arpit et al. (2017); Sanyal et al. (2021), which also relates our studies. Not less interesting, the learning risk of highly-overparameterized models shows a *double-descent* behavior when varying the model

complexity Nakkiran et al. (2020) as yet another phenomenon. Lastly, the concept of NC Papyan et al. (2020) has recently shed light on symmetries in learned representations of overparameterized models.

After laying the foundation of a rigorous mathematical characterization of the NC phenomenon by Papyan et al. (2020), several follow-up works have broadened the picture. As the former proceeds from studying CE loss, the collapsing behavior has been investigated for alternative loss functions. For instance, squared losses have shown similar collapsing characteristics Poggio & Liao (2020; 2021), and have paved the way for more opportunities in its mathematical analysis, e.g., by an NC-interpretable decomposition Han et al. (2021). More recently, Kornblith et al. (2021) provide an exhaustive overview over several commonly used loss functions for training deep neural networks regarding their feature collapses.

Besides varying the loss function, different theoretical models have been proposed to analyze NC. Most prominently, *unconstrained feature models* have been considered, which characterize the penultimate layer activations as free optimization variables Mixon et al. (2020); Lu & Steinerberger (2020); E & Wojtowytsch (2021). This stems from the assumption that highly overparameterized models can approximate any patterns in the feature space. While unconstrained features models typically only look at the last feature encoder layer, *layer-peeling* allows for "white-boxing" further layers before the last one for a more comprehensive theoretical analysis Fang et al. (2021). Indeed, this approach has been applied in Tirer & Bruna (2022), which namely extends the unconstrained features model by one layer as well as the ReLU nonlinearity. On the other hand, Zhu et al. (2021), Ji et al. (2021) and Zhou et al. (2022a) extend the unconstrained features model analysis by studying the landscape of the loss function therein and the related training dynamics. Beyond unconstrained features models, Ergen & Pilanci (2021) introduce a convex analytical framework to characterize the encoder layers for a more profound understanding of the NC phenomenon. Referring to the implications of NC on our understanding of neural networks, Hui et al. (2022) and Galanti et al. (2021) discuss the impact of NC on test data in the sense of generalization and transfer learning. Finally, Kothapalli et al. (2022) provides a multifaceted survey of recent works related to NC.

## 3 LAYER-PEELED MODEL WITH POSITIVE FEATURES

As a prerequisite to the MD model, in this section we introduce a slightly modified version of the layer-peeled (or unconstrained features) model (see e.g. Zhu et al. (2021); Fang et al. (2021)), in which the features have to be positive. Accordingly, we will show that the global minimizers of the modified layer-peeled model correspond to an NC configuration, which differs from the global minimizers specified in other works and captures more closely the NC phenomenon in practice.

For conciseness, we denote by $\boldsymbol{H}$ the matrix formed by the features $\boldsymbol{h}_n^{(k)}$, $n \in [N]$, $k \in [K]$ as columns, and define $\|\boldsymbol{W}\|$ and $\|\boldsymbol{H}\|$ to be the Frobenius norm of the respective matrices, i.e. $\|\boldsymbol{W}\|^2 = \sum_{n=1}^N \|\boldsymbol{w}_n\|^2$ and $\|\boldsymbol{H}\|^2 = \sum_{k=1}^K \sum_{n=1}^N \left\|\boldsymbol{h}_n^{(k)}\right\|^2$. We consider the regularized version of the model (instead of the norm constraint one as in e.g. Fang et al. (2021)) [1]

$$\min_{\boldsymbol{W},\boldsymbol{H}} \quad \mathcal{L}_\alpha(\boldsymbol{W},\boldsymbol{H}) := L_\alpha(\boldsymbol{W},\boldsymbol{H}) + \lambda_W \|\boldsymbol{W}\|^2 + \frac{\lambda_H}{K} \|\boldsymbol{H}\|^2 \qquad (\mathcal{P}_\alpha)$$
$$\text{s.t.} \quad \boldsymbol{H} \geq 0,$$

where $\lambda_W, \lambda_H > 0$ are the penalty parameters for the weight decays. By $L_\alpha$ we denote empirical risk with respect to the LS loss with parameter $\alpha \in [0,1)$, where $\alpha = 0$ corresponds to the conventional CE loss. More precisely, given a value of $\alpha$, the LS technique then defines the label assigned to class $n \in [N]$ as the following probability vector:

$$\boldsymbol{y}_n^{(\alpha)} = (1-\alpha)\boldsymbol{e}_n + \frac{\alpha}{n}\mathbf{1}_N \in [0,1]^N,$$

where $\boldsymbol{e}_n \in \mathbb{R}^N$ denotes the $n$-th standard basis vector and $\mathbf{1}_N \in \mathbb{R}^N$ denotes the vector consisting of only ones. Let $p : \mathbb{R}^M \to \mathbb{R}^N$ be the function that assigns to each feature representation $\boldsymbol{z} \in \mathbb{R}^M$

---

[1]Note that for simplicity we assume that the last layer does not have bias terms, i.e. $b = 0$. The result can be however easily extended to the more general case when the biases do not vanish. Namely, in presence of bias terms, the statement of Theorem 3.2 and also its proof remain unchanged.

the probability scores of the classes (as a probability vector in $\mathbb{R}^N$),

$$p_{\boldsymbol{W}}(\boldsymbol{z}) := \mathrm{softmax}(\boldsymbol{W}\boldsymbol{z}) := \left[ \frac{e^{\langle \boldsymbol{w}_m, \boldsymbol{z} \rangle}}{\sum_{i=1}^N e^{\langle \boldsymbol{w}_i, \boldsymbol{z} \rangle}} \right]_{m=1}^N \in [0,1]^N.$$

Then the LS loss corresponding to a sample in class $n \in [N]$ is given by

$$\ell_\alpha(\boldsymbol{W}, \boldsymbol{z}, \boldsymbol{y}_n^{(\alpha)}) := \left\langle -\boldsymbol{y}_n^{(\alpha)}, \log p_{\boldsymbol{W}}(\boldsymbol{z}) \right\rangle := \sum_{m=1}^N -\boldsymbol{y}_{nm}^{(\alpha)} \log \left( p_{\boldsymbol{W}}(\boldsymbol{z})_m \right) \tag{1}$$

and the LS empirical risk $L_\alpha$ is defined as

$$L_\alpha(\boldsymbol{W}, \boldsymbol{H}) = \frac{1}{NK} \sum_{k=1}^K \sum_{n=1}^N \ell_\alpha \left( \boldsymbol{W}, \boldsymbol{h}_n^{(k)}, \boldsymbol{y}_n^{(\alpha)} \right).$$

We will show that in common settings, the minimizers of $(\mathcal{P}_\alpha)$ correspond to *neural collapse (NC) configurations*, which we formalize in Def. 3.1 below.

**Definition 3.1** (NC configurations). Let $K, M, N \in \mathbb{N}$, $M \geq N$. A pair $(\boldsymbol{W}, \boldsymbol{H})$ of a weight matrix formed by rows $\boldsymbol{w}_n \in \mathbb{R}^M$ and a feature matrix formed by columns $\boldsymbol{h}_n^{(k)} \in \mathbb{R}_+^M$ (with $n \in [N], k \in [K]$) is said to be a *NC configuration* if

(i) The feature representations $\boldsymbol{h}_n^{(k)}$ within every class $n \in [N]$ are equal for all $k \in [K]$, and thus equal to their class mean $\boldsymbol{h}_n := \frac{1}{K} \sum_{k=1}^K \boldsymbol{h}_n^{(k)}$.

(ii) The class means $\{\boldsymbol{h}_n\}_{n=1}^N$ have equal norms and form an (entry-wise) non-negative orthogonal system.

(iii) Let $P_{\boldsymbol{h}^\perp}$ be the projection upon the subspace of $\mathbb{R}^M$ orthogonal to $\boldsymbol{h} = \frac{1}{N} \sum_{n=1}^N h_n$. Then for every $n \in [N]$, it holds $\boldsymbol{w}_n = C P_{\boldsymbol{h}^\perp} \boldsymbol{h}_n$ for some constant $C$ independent of $n$.

Our main theorem in this section can be represented as follows.

**Theorem 3.2.** *Let $M \geq N$, $\alpha \in [0,1)$. Assume that $\frac{N-1}{N}\alpha + 2\sqrt{(N-1)\lambda_W \lambda_H} < 1$. Then any global minimizer of the problem $(\mathcal{P}_\alpha)$ is a NC configuration.*

Note that the NC configurations defined in Definition 3.1 above differ significantly from the ones specified in other works, e.g. Fang et al. (2021); Zhu et al. (2021); Zhou et al. (2022b) or Tirer & Bruna (2022), see Appendix B.1 for more discussion.

# 4 THE MEMORIZATION-DILATION MODEL

## 4.1 EXPERIMENTAL MOTIVATION

Previous studies of the NC phenomenon mainly focus on the collapsing variability of *training* activations, and make rather cautious statements about its effects on generalization. For instance, Papyan et al. (2020) report slightly improved test accuracies for training beyond zero training error. Going a step further, Zhu et al. (2021) show that the NC phenomenon also happens for overparameterized models when labels are completely randomized. Here, the models seem to *memorize* by overfitting the data points, however, a rigorous study how label corruption affects generalization in the regime of NC is still lacking.

To fill the gap, we advocate to analyze the effects of label corruption in the training data on the (previously unseen) *test* instead of the training feature collapse. Eventually, tight test class clusters go hand in hand with easier separation of the instances and, thus, a smaller generalization error. Following Zhu et al. (2021), we measure the collapse of the penultimate layer activations by the $\mathcal{NC}_1$ metric. This metric depicts the relative magnitude of the within-class covariance $\boldsymbol{\Sigma}_W$ with respect to the between-class covariance $\boldsymbol{\Sigma}_B$ of the penultimate layer features and is defined as

$$\mathcal{NC}_1 := \frac{1}{N} \mathrm{trace}(\boldsymbol{\Sigma}_W \boldsymbol{\Sigma}_B^\dagger), \tag{2}$$

where

$$\mathbf{\Sigma}_W := \frac{1}{NK} \sum_{n=1}^{N} \sum_{k=1}^{K} (\boldsymbol{h}_n^{(k)} - \boldsymbol{h}_n)(\boldsymbol{h}_n^{(k)} - \boldsymbol{h}_n)^\top \in \mathbb{R}^{M \times M} \,,$$

$$\mathbf{\Sigma}_B := \frac{1}{N} \sum_{n=1}^{N} (\boldsymbol{h}_n - \boldsymbol{h})(\boldsymbol{h}_n - \boldsymbol{h})^\top \in \mathbb{R}^{M \times M},$$

and $\mathbf{\Sigma}_B^\dagger$ denotes the pseudo-inverse of $\mathbf{\Sigma}_B$. Here, we adopt the notations from Section 1: $\boldsymbol{h}_n^{(k)} \in \mathbb{R}^M$ denotes the feature representation of $k$-th sample in class $n$, $\boldsymbol{h}_n$ the class mean and $\boldsymbol{h}$ the global mean. Moreover, we distinguish $\mathcal{NC}_1^{\text{train}}$ and $\mathcal{NC}_1^{\text{test}}$ to be calculated on the training and test instances, respectively. We call $\mathcal{NC}_1^{\text{test}}$ *dilation*.

Let us now turn to the notion of *memorization*, which is not uniquely defined in deep learning literature. Here, we define memorization in the context of the NC setting and in a global manner, different from other works, e.g. Feldman & Zhang (2020). Formally, suppose that label noise is incorporated by (independently) corrupting the instance of each class label $n$ in the training data with probability $\eta \in (0, 1)$, where corruption means drawing a label uniformly at random from the label space $\mathcal{Y}$. We denote the set of corrupted instances by $[\widetilde{K}]$. For a given dataset $\mathcal{D}$ (with label noise $\eta$), we define *memorization* as

$$\text{mem} := \sum_{n=1}^{N} \sum_{k \in [\widetilde{K}]} \|\boldsymbol{h}_n^{(k)} - \boldsymbol{h}_n^*\|_2 \,, \tag{3}$$

where $\boldsymbol{h}_n^*$ denotes the mean of (unseen) test instances belonging to class $n$.

We call the original ground truth label of a sample its *true label*. We call the label after corruption, which may be the true label or not, the *observed label*. Since instances of the same true label tend to have similar input features in some sense, the network is biased to map them to similar feature representations. Instances are corrupted randomly, and hence, instances of the same true label but different observed labels do not have predictable characteristics that allow the network to separate them in a way that can be generalized. When the network nevertheless succeeds in separating such instances, we say that the network *memorized* the feature representations of the corrupted instances in the training set. The metric mem in (3) thus measures memorization. The above memorization also affects dilation. Indeed, the network uses the feature engineering part to embed samples of similar features (that originally came from the same class), to far apart features, that encode different labels. Such process degrades the ability of the network to embed samples consistently, and leads to dilation.

To quantify the interaction between mem and $\mathcal{NC}_1^{\text{test}}$, we analyzed the learned representations $\boldsymbol{h}$ in the penultimate layer feature space for different noise configurations. One may wonder whether one can see a systematic trend in the test collapse given the memorization, and how this evolves over different loss functions.

To this end, we trained simple multi-layer neural networks for two classes ($N = 2$), which we subsampled from the image classification datasets MNIST LeCun et al. (1998), FashionMNIST Xiao et al. (2017), CIFAR-10 Krizhevsky & Hinton (2009) and SVHN Netzer et al. (2011). The labels are corrupted with noise degrees $\eta \in [0.025, 0.4]$. The network consists of 9 hidden layers with 2048 neurons each, thus, it represents a vastly overparameterized model. The feature dimension $M$ is set to the number of classes $N$. We trained these networks using the CE and LS loss with a smoothing factor $\alpha = 0.1$, as well as the mean-squared error (MSE). Moreover, we consider label relaxation (LR) Lienen & Hüllermeier (2021) as a generalization to LS with a relaxation degree $\alpha = 0.1$. The networks were trained until convergence in 200 epochs (where the last 50 epochs did not make any significant changes) using SGD with an initial learning rate of 0.1 multiplied by 0.1 each 40 epochs and a small weight decay of 0.001. Moreover, we considered ReLU as activation function throughout the network, as well as batch normalization in each hidden layer. A linear softmax classifier is composed on the encoder. We conducted each experiment ten times with different seeds.

The results for the above experimental setting are shown in Fig. 1, in which one can observe the trends of $\sqrt{\mathcal{NC}_1^{\text{test}}}$ per memorization for various configurations. As can be seen, the figure shows an approximately linear correspondence between $\sqrt{\mathcal{NC}_1^{\text{test}}}$ and mem for the CE derivatives (CE and LS) on all datasets when mem is not large.

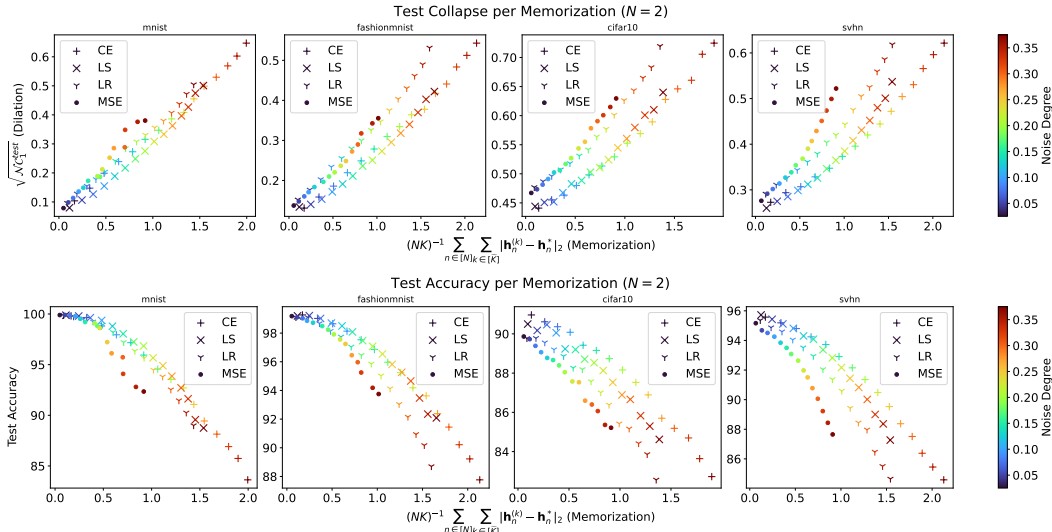

Figure 1: Feature collapse of the test instances in terms of $\sqrt{\mathcal{NC}_1^{\text{test}}}$ per memorization (top row) and the resulting test accuracies (bottom row) averaged over ten random seeds. Comparing the markers of the same color, it can be observed that LS consistently performs better than CE across all datasets, with very few exceptions (the very low noise degrees in cifar10).

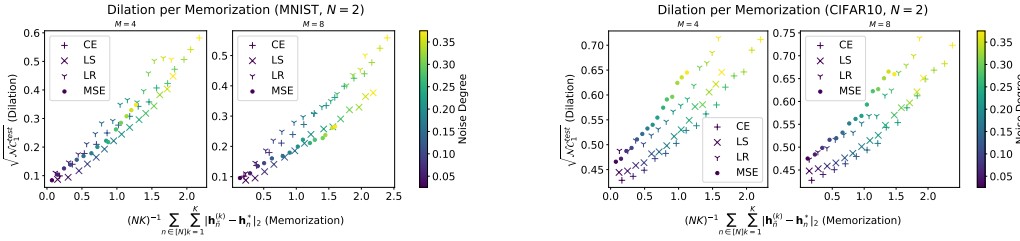

Figure 2: Feature collapse of the test instances in terms of $\sqrt{\mathcal{NC}_1^{\text{test}}}$ per memorization as the feature dimension $M$ varies.

Moreover, as CE and LS share the same slope, these results suggest that the degradation of the test collapse (aka dilation) is a function of memorization and the network expressitivity, and not of the choice of the loss. The loss only affects how the noise translates to memorization, but not how memorization translates to dilation. Even though the same amount of noise is mapped to different memorization values in CE and LS, the memorization-dilation curve is nevertheless shared between CE and LS. Hence, since LS leads the network to memorize less, it results in improved performance (cf. Fig. 1). We can further see that MSE and LR show a different memorization-dilation correspondence, which means that these losses affect the inductive bias in a different way than CE and LS.

We repeated the experiments for different values of the feature dimension $M$ and show the example results in Fig. 2. Here, one can see the similar trends of dilation per memorization as before. In the appendix, we provide additional results showing the behavior in the multi-class case $N > 2$ with different models for label noise. The results support our MD model, and show that the memorization-dilation curve is roughly independent of the noise model for low-to-mid noise levels.

## 4.2 THE MEMORIZATION-DILATION MODEL

Motivated by the observations of the previous experiments, we propose the so-called *memorization-dilation (MD) model*, which extends the unconstrained feature model by incorporating the interaction between memorization and dilation as a model assumption. By this, we explicitly capture the limited expressivity of the network, thereby modeling the inductive bias of the underlying model.

This model shall provide a basis to mathematically characterize the difference in the learning behavior of CE and LS. More specifically, we would like to know *why* LS shows improved generalization performance over conventional CE, as was observed in past works Müller et al. (2019). The main idea can be explained as follows. We first note that dilation is directly linked to generalization (see also Kornblith et al. (2021)), since the more concentrated the feature representations of each class are, the easier it is to separate the different classes with a linear classifier without having outliers crossing the decision boundary. The MD model asserts that dilation is a linear function of memorization. Hence, the only way that LS can lead to less dilation than CE, is if LS memorizes less than CE. Hence, the goal in our analysis is to show that, under the MD model, LS indeed leads to less memorization than CE. Note that this description is observed empirically in the experiments of Section 4.1.

Next we define the MD model in the binary classification setting.

**Definition 4.1.** We call the following minimization problem $\mathcal{MD}$. Minimize the *MD risk*

$$\mathcal{R}_{\lambda,\eta,\alpha}(\boldsymbol{U}, r) := F_{\lambda,\alpha}(\boldsymbol{W}, \boldsymbol{H}, r) + \eta G_{\lambda,\alpha}(\boldsymbol{W}, \boldsymbol{U}, r),$$

with respect to the *noisy feature embedding* $\boldsymbol{U} = [\boldsymbol{u_1}, \boldsymbol{u_2}] \in \mathbb{R}_+^{2 \times M}$ and the *standard deviation* $r \geq 0$, under the constraints

$$\eta \|\boldsymbol{h}_1 - \boldsymbol{u}_2\| \leq \frac{C_{MD} r}{\|\boldsymbol{h}_1 - \boldsymbol{h}_2\|} \tag{4}$$

$$\eta \|\boldsymbol{h}_2 - \boldsymbol{u}_1\| \leq \frac{C_{MD} r}{\|\boldsymbol{h}_1 - \boldsymbol{h}_2\|}. \tag{5}$$

Here,

- $\boldsymbol{H} \in \mathbb{R}_+^{2 \times M}$ and $\boldsymbol{W} \in \mathbb{R}^{M \times 2}$ form an NC configuration (see Definition 3.1).

- $C_{MD} > 0$ is called the *memorization-dilation slope*, $0 \leq \alpha < 1$ is called the *LS parameter*, $\eta > 0$ the *noise level*, and $\lambda > 0$ the *regularization parameter*.

- $F_{\lambda,\alpha}$ is the component in the (regularized) risk that is associated with the correctly labeled samples,

$$F_{\lambda,\alpha}(\boldsymbol{W}, \boldsymbol{H}, r) := \int \left( \ell_\alpha \Big( \boldsymbol{W}, \boldsymbol{h}_1 + \boldsymbol{v}, \boldsymbol{y}_1^{(\alpha)} \Big) + \lambda \|\boldsymbol{h}_1 + \boldsymbol{v}\|^2 \right) d\mu_r^1(\boldsymbol{v})$$

$$+ \int \left( \ell_\alpha \Big( \boldsymbol{W}, \boldsymbol{h}_2 + \boldsymbol{v}, \boldsymbol{y}_2^{(\alpha)} \Big) + \lambda \|\boldsymbol{h}_2 + \boldsymbol{v}\|^2 \right) d\mu_r^2(\boldsymbol{v})$$

  where $\mu_r^1$ and $\mu_r^2$ are some probability distributions with mean $0$ and standard deviation $r$, and $l_\alpha$ is the LS loss defined in (1).

- $G_{\lambda,\alpha}$ is the component in the (regularized) risk that is associated with the corrupted samples, defined as

$$G_{\lambda,\alpha}(\boldsymbol{W}, \boldsymbol{U}, r) = \ell_\alpha \Big( \boldsymbol{W}, \boldsymbol{u_1}, \boldsymbol{y}_1^{(\alpha)} \Big) + \ell_\alpha \Big( \boldsymbol{W}, \boldsymbol{u_2}, \boldsymbol{y}_2^{(\alpha)} \Big) + \lambda \|\boldsymbol{u_1}\|^2 + \lambda \|\boldsymbol{u_2}\|^2.$$

The MD model can be interpreted as follows. First we consider the feature representations of the correctly labeled samples in each class as samples from a distribution (namely $\mu_r^{1,2}$ in Def. 4.1) with standard deviation $r$, a parameter that measures the dilation of the class cluster. In a natural way, the corresponding risk $F_{\lambda,\alpha}$ involves the loss average over all samples, i.e. the loss integral over the distribution. For simplicity, we assume that the class centers $\boldsymbol{h}_1, \boldsymbol{h}_2$ as well as the weight matrix $\boldsymbol{W}$ are fixed as described by the NC configuration. This is a reasonable simplification as it has been always observed in the experiments.

On the other hand, the feature representations of corrupted samples are $\boldsymbol{u}_1$ and $\boldsymbol{u}_2$.[2] The amount of memorization in the first class is defined to be $\eta \|\boldsymbol{h}_2 - \boldsymbol{u}_1\|$, since the more noise $\eta$ there is, the more

---

[2]Certainly one can, instead of two single points $\boldsymbol{u}_1$ and $\boldsymbol{u}_2$, two distributions centered around $\boldsymbol{u}_1$ and $\boldsymbol{u}_2$, similarly as before for uncorrupted samples. However, it is quite straightforward to see that the minimization of the MD risk over the dilation of these two distributions is independent of other variables (not like $r$), and thus the minimum should be attained in the case of collapsing into two single points. Thus, for convenience we assume directly here that $G_{\lambda,\alpha}$ involves only two single points.

examples we need to memorize. The amount of memorization in the second class is defined the same way. The (normalized) dilation is defined to be $\frac{r}{\|\boldsymbol{h}_1 - \boldsymbol{h}_2\|}$, which models a similar quantity to (2).

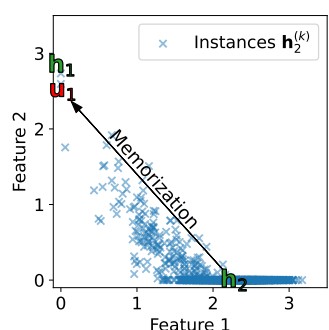

The constraints (4) and (5) tell us that in order to map noisy samples $\boldsymbol{u}_1$ away from $\boldsymbol{h}_2$, we have to pay with dilation $r$. The larger $r$ is, the further away we can map $\boldsymbol{u}_1$ from $\boldsymbol{h}_2$. The correspondence between memorization and dilation is linear with slope $C_{MD}$ by assumption. There are two main forces in the optimization problem: $\boldsymbol{u}_1$ would like to be as close as possible to its optimal position $\boldsymbol{h}_1$, and similarly $\boldsymbol{u}_2$ likes to be close to $\boldsymbol{h}_2$. In view of the constraints (4) and (5), to achieve this, $r$ has to be increased to $r_{\max} := \frac{\eta \|\boldsymbol{h}_1 - \boldsymbol{h}_2\|^2}{C_{MD}}$. On the other hand, the optimal $r$ for the term $F_{\lambda, \alpha}$ is $r = 0$, namely, the layer-peeled NC configuration. An optimal solution hence balances between memorization and dilation. See Fig. 3 for a visualization of the MD model.

Our goal in this section is to compare the optimal value $r$ in case of LS and CE losses. We will distinguish between these two cases by setting the value of $\alpha$ in the MD model to $0$ for CE and to some $\alpha_0 > 0$ for LS. This will result in two different scales of the feature embeddings $\boldsymbol{H}$, denoted by $\boldsymbol{H}^{CE}$ and $\boldsymbol{H}^{LS}$ for CE and LS loss respectively, with the ratio

Figure 3: Exemplary illustration of the MD model for a MLP network trained on MNIST. The instances $\boldsymbol{h}_2^{(k)}$ are test images correctly labeled as 1, with centroid $\boldsymbol{h}_2$. The centroid of the test images with correct label 0 is $\boldsymbol{h}_1$. The centroid of training images which were originally labeled as 1 but are mislabeled as 0 is $\boldsymbol{u}_1$. The memorization of $\boldsymbol{u}_1$ moves it close to $\boldsymbol{h}_1$, and causes dilation of the instances $\boldsymbol{h}_2^{(k)}$.

$$\gamma := \left\| \boldsymbol{H}^{CE} \right\| / \left\| \boldsymbol{H}^{LS} \right\| > 1, \tag{6}$$

which holds under the reasonable assumption that the LS technique is sufficiently effective, or more precisely $\alpha_0 > 2\sqrt{\lambda_W \lambda_H}$.

The main result in this section will be Theorem 4.3, which states informally that in the low noise regime, the optimal dilation in case of LS loss is smaller than that in case of CE loss. Before presenting this theorem, we will first establish several assumptions on the distributions $\mu_r^{1,2}$ and the noise $\eta$ in Assumption 4.2. Basically we allow a rich class of distributions and only require certain symmetry and bounded supports in terms of $r$, as well as require $\eta$ to be small in terms of the ratio $\gamma$.

**Assumption 4.2.**

1. Let $\alpha_0 > 0$. We assume that the solution of

$$\min_{\boldsymbol{W}, \boldsymbol{H}} \quad \ell_\alpha \left( \boldsymbol{W}, \boldsymbol{h}_1, \boldsymbol{y}_1^{(\alpha)} \right) + \ell_\alpha \left( \boldsymbol{W}, \boldsymbol{h_2}, \boldsymbol{y_2}^{(\alpha)} \right) + \lambda_W \left\| \boldsymbol{W} \right\|^2 + \lambda_H \left\| \boldsymbol{H} \right\|^2$$
$$\text{s.t.} \quad \boldsymbol{H} \geq 0.$$

   is given by $(\boldsymbol{W}, \boldsymbol{H}) = (\boldsymbol{W}^{CE}, \boldsymbol{H}^{CE})$ for $\alpha = 0$ and $(\boldsymbol{W}, \boldsymbol{H}) = (\boldsymbol{W}^{LS}, \boldsymbol{H}^{LS})$ for $\alpha = \alpha_0$.

2. Assume that the distributions $\mu_r^1$ and $\mu_r^2$ are *centered*, in the sense that

$$\int \langle \boldsymbol{w}_2 - \boldsymbol{w}_1, \boldsymbol{v} \rangle \, d\mu_r^1(\boldsymbol{v}) = \int \langle \boldsymbol{w}_1 - \boldsymbol{w}_2, \boldsymbol{v} \rangle \, d\mu_r^2(\boldsymbol{v}) = 0 \,,$$

$$\int \langle \boldsymbol{h}_1, \boldsymbol{v} \rangle \, d\mu_r^1(\boldsymbol{v}) = \int \langle \boldsymbol{h}_2, \boldsymbol{v} \rangle \, d\mu_r^2(\boldsymbol{v}) = 0 \,.$$

   Furthermore, we assume that there exists a constant $A > 0$ such that $\|\boldsymbol{v}\| \leq Ar$ for any vector $\boldsymbol{v}$ that lies in the support of $\mu_r^1$ or in the support of $\mu_r^2$.

3. Assume that the noise level $\eta$ and the LS parameter $\alpha_0$ satisfy the following. We suppose $\alpha_0 > 4\sqrt{\lambda_W \lambda_H}$, which guarantees $\gamma := \left\| \boldsymbol{H}^{CE} \right\| / \left\| \boldsymbol{H}^{LS} \right\| > 1$. We moreover suppose that $\eta$ is sufficiently small to guarantee $\eta^{1/2} < \tilde{C}(1 - \frac{1}{\gamma})$ where $\tilde{C} := \frac{C_{MD}}{\sqrt{2} \| \boldsymbol{h}_1^{CE} - \boldsymbol{h}_2^{CE} \|}$.

Now our main result in this section can be formally stated as below.

**Theorem 4.3.** *Suppose that Assumption 4.2 holds true for $M \geq N = 2$ and $\lambda := \lambda_H$. Let $r_*^{CE}$ and $r_*^{LS}$ be the optimal dilations, i.e. the optimum $r$ in the $\mathcal{MD}$ problem, corresponding to the CE and LS loss (accordingly $\alpha = 0$ and $\alpha = \alpha_0$), respectively. Then it holds that*

$$\frac{r_*^{CE}}{\left\| \boldsymbol{h}_1^{CE} - \boldsymbol{h}_2^{CE} \right\|} > \frac{r_*^{LS}}{\left\| \boldsymbol{h}_1^{LS} - \boldsymbol{h}_2^{LS} \right\|} \, .$$

Theorem 4.3 reveals a mechanism by which LS achieves better generalization than CE. It is proven that LS memorizes and dilates less than CE, which is associated with better generalization. Note that in practice, the data often have noise in the sense that not all examples are perfectly labeled. More importantly, examples from different classes may share many similarities, a situation that is also covered by the MD model: the feature representations of samples from those classes are biased toward each other. In this case, LS also leads to decreased dilation which corresponds to better class separation and higher performance Kornblith et al. (2021).

Interestingly, the concurrent work Zhou et al. (2022b) has shown that in the noiseless setting CE and LS lead to largely identical test accuracy, which seems to contradict the statement that LS performs better claimed by our work as well as many others, e.g. Kornblith et al. (2021); Müller et al. (2019). However, note that Zhou et al. (2022b) requires the network to be sufficiently large so that it has enough expressive power to fit the underlying mapping from input data to targets, as well as to be trained until convergence. While the latter is easy to obtain, it is difficult even to check if the first requirement holds. The difference between the two results is hence possibly caused by the effect of noise and by the network expressivity: while we aim to model the limited expressivity by the MD relation, Zhou et al. (2022b) focuses on networks with approximately infinite expressivity.

The MD model combines a statistical term $F_{\lambda,\alpha}$, that describes the risk over the distribution of feature embeddings of samples with clean labels, and an empirical term $\eta G_{\lambda,\alpha}$ that describes the risk over training samples with noisy labels. One point of view that can motivate such a hybrid statistical-empirical definition is the assumption that the network only memorizes samples of noisy labels, but not samples of clean labels. Such a memorization degrades (dilates) both the collapse of the training and test samples, possibly with different memorization-dilation slopes. However, memorization is not limited to corrupted labels, but can also apply to samples of clean labels Feldman & Zhang (2020), by which the learner can partially negate the dilation of the training features (but not test features). The fact that our model does not take the memorization of clean samples into account is one of its limitations. We believe that future work should focus on modeling the total memorization of all examples. Nevertheless, we believe that our current MD model has merit, since 1) noisy labels are memorized more than clean labels, and especially in the low noise regime the assumption of observing memorization merely for corrupted labels appears reasonable, and 2) our approach and proof techniques can be the basis of more elaborate future MD models.

## 5 CONCLUSION

In this paper, we first characterized the global minimizers of the Layer-Peeled Model (or the Unconstrained Features Model) with the positivity condition on the feature representations. Our characterization shows some distinctions from the results that haven been obtained in recent works for the same model without feature positivity. Besides the conventional cross-entropy (CE) loss, we studied the model in case of the label smoothing (LS) loss, showing that NC also occurs when applying this technique.

Then we extended the model to the so-called Memorization-Dilation (MD) Model by incorporating the limited expressivity of the network. Using the MD model, which is supported by our experimental observations, we show that when trained with the LS loss, the network memorizes less than when trained by the CE loss. This poses one explanation to the improved generalization performance of the LS technique over the conventional CE loss.

Our model has limitations, however, namely that it is limited to the case of two classes. Motivated by promising results on the applicability of our model to the multi-class setting, we believe that future work should focus on extending the MD model in this respect. With such extensions, memorization-dilation analysis has the potential to underlie a systematic comparison of the generalization capabilities of different losses, such as CE, LS, and label relaxation, by analytically deriving formulas for the amount of memorization associated with each loss.

ACKNOWLEDGMENTS

This work was partially supported by the German Research Foundation (DFG) within the Collaborative Research Center "On-The-Fly Computing" (CRC 901 project no. 160364472). Moreover, the authors gratefully acknowledge the funding of this project by computing time provided by the Paderborn Center for Parallel Computing ($PC^2$).

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
