# OpenReview forum: "Memorization-Dilation: Modeling Neural Collapse Under Noise"
_ICLR.cc/2023/Conference — ICLR 2023 poster_

### Official Review · Reviewer_xSrW · 2022-10-24

**Confidence:** 3
**Correctness:** 3
**Technical Novelty And Significance:** 3
**Empirical Novelty And Significance:** 2
**Recommendation:** 6

**Clarity, Quality, Novelty And Reproducibility:**

The presentation of this paper is relatively easy to follow. I do have a nitpick request: I think a more common convention is to use n, N for example indices and k, K for class indices. This paper does the opposite and confused me a number of times during the reading.

**Strength And Weaknesses:**

**Strength**
1. This paper theoretically studied neural collapse (NC) with a definition that describes more precisely the empirically observed phenomena in practice comparing to other papers.
2. This paper formally study the NC under label noises with a memorization-dilation model, and make connection between memorization and NC.
3. The theoretical model is motivated from empirical observations with deep neural networks on real data.

**Weakness**
1. The theoretical setup for studying neural collapse is quite different from what a real neural network behave in practice. In particular, it assumes "infinite expressivity" and allow the representation of each example to be freely trainable parameters. The resulting formulation looks like a matrix factorization problem with a classification loss. I think the most interesting part of neural network representation learning (and collapsing) would mostly happen jointly in the layers below the final classifier layer, and how those layers share weights when jointly computing the representations for all the training examples. This is especially important for the topic of this paper, where label noise is introduced to study memorization. In this case, the lower layers are forced to build different representations for visually similar inputs in some cases when they are assigned different training labels. But this interactions would be completely missing in the model proposed here.

2. I appreciate that the empirical studies in this paper uses deep neural networks. However, since the theoretical models are so different, I would like to see empirical studies with a similar setup, using networks with approximately infinite expressivity. Or better, simply using a model where the representations are directly optimizable free variables, and see if the empirical observations are still similar and equally well for motivating the M-D models. One question here is how to get the representations for the test examples for computing Eq (3) in this case.

3. The analysis is limited to binary classification problems.

**Summary Of The Paper:**

This paper theoretically studies the notion of neural collapse (NC) -- the penultimate layer representations of all the examples in a specific class collapse to a single representation, and separate from other classes. Specifically, this paper studies this notion under uniform label noises, and proposed a model called memorization-dilation (M-D), in which it shows labels smoothing leads to less memorization and better generalization.

**Summary Of The Review:**

This paper studies the phenomenon of neural collapse (NC) under label noises using a theoretical model to characterize the relation between memorization and NC. One of my major concern is that the theoretical model is too different from a real deep neural network, which is especially crucial for the topic of this paper (label noises and memorization).

-------
Post rebuttal: thanks to the authors for clarifying my questions.

---

> ### Author Response · Authors · 2022-11-18
> **Response to reviewer xSrW (1/2)**
>
> We are very grateful to reviewer xSrW for the time taken to carefully assess our work and for the valuable feedback. Below, we address each point individually. “R” quotes the Reviewer and “A” marks the response by the authors.
>
> > **R**: *"The theoretical setup for studying neural collapse is quite different from what a real neural network behave in practice. In particular, it assumes "infinite expressivity" and allow the representation of each example to be freely trainable parameters. The resulting formulation looks like a matrix factorization problem with a classification loss. I think the most interesting part of neural network representation learning (and collapsing) would mostly happen jointly in the layers below the final classifier layer, and how those layers share weights when jointly computing the representations for all the training examples. This is especially important for the topic of this paper, where label noise is introduced to study memorization. In this case, the lower layers are forced to build different representations for visually similar inputs in some cases when they are assigned different training labels. But this interactions would be completely missing in the model proposed here."*
>
> **A**: Reply: To be clear, we study two different models: the layer-peeled model (with positive features) assumes infinite expressivity of the network while the memorization-dilation model does not. The point of the first model (the layer peeled model) is just to find an appropriate configuration as a basis of our second model, which studies networks with limited expressivity. The second model (the MD model) is the focus of our paper. The point in this paper is to enhance the layer peeled model, which assumes infinite expressivity, by modeling into it the finite expressivity of the neural network. We do that by taking the MD assumptions.
> The point about the lower layers (that we call the feature engineering part) is valid. In fact, this idea is implicitly modeled and is one of the motivations of our memorization-dilation model. This was not explained well enough in the original paper, and we hence revised the explanation as follows.
>
> First in Page 5 (that was also in the original version):
>
> “We call the original ground truth label of a sample its \emph{true label}. We call the label after corruption, which may be the true label or not, the \emph{observed label}. Since instances of the same true label tend to have similar input features in some sense, the network is biased to map them to similar feature representations. Instances are corrupted randomly, and hence, instances of the same true label but different observed labels do not have predictable characteristics that allow the network to separate them in a way that can be generalized. When the network nevertheless succeeds in separating such instances, we say that the network \emph{memorized} the feature representations of the corrupted instances in the training set. The metric $\text{mem}$ in (3) thus measures memorization. “
>
> “To quantify the interaction between $\text{mem}$ and $\mathcal{NC}_1^{\text{test}}$, we analyzed the learned representations ${h}$ in the penultimate layer feature space for different noise configurations. One may wonder whether one can see a systematic trend in the test collapse given the memorization, and how this evolves over different loss functions.”
>
> We then added the following new text:
>
> “The above memorization also affects dilation. Indeed, the network uses the feature engineering part to embed samples of similar features (that originally came from the same class), to far apart features, that encode different labels. Such a process degrades the ability of the network to embed samples consistently, and leads to dilation.”

---

> > ### Author Response · Authors · 2022-11-18
> > **Response to reviewer xRsW (2/2)**
> >
> > > **R**: *"I appreciate that the empirical studies in this paper uses deep neural networks. However, since the theoretical models are so different, I would like to see empirical studies with a similar setup, using networks with approximately infinite expressivity. Or better, simply using a model where the representations are directly optimizable free variables, and see if the empirical observations are still similar and equally well for motivating the M-D models. One question here is how to get the representations for the test examples for computing Eq (3) in this case."*
> >
> > **A**: As explained above, we did not assume infinite expressivity in the MD model. Indeed, we aim to study networks with limited expressivity and hence did not perform the experiments for huge networks suggested by the reviewer. However, below we will provide a few thoughts on the suggested model where the features are directly optimizable free variables.
> > When using models where the features are directly optimizable free variables, the empirical results simply suggest zero dilation (with the constraints (4) and (5)), i.e. perfect collapse of all features in each class. This is not surprising because the infinite expressivity allows the feature representations to collapse to any point, while the underlying minimization problem restricted to each feature vector is convex. We did perform such experiments that gave these exact results, but since such results are trivial, we did not include them in the paper.
> >
> > >**R**: *"The analysis is limited to binary classification problems."*
> >
> > **A**: In general, we aimed for a theoretical work with rigorous proofs, and hence we had to restrict ourselves to a simplified setting. We expect our work to be the first step in a sequence of papers, by ourselves and others, that will build upon each other and make the learning scenario more and more realistic. The study of binary classification problems is the starting point, based on which one could further explore the more general cases, including e.g. multi-class or the data imbalance setting. In fact, the experimental results for the multi-class case look promising (see Figure 4,5,6 in the appendix), however we could not explain them rigorously as in the binary case and had to leave this for future research. Only rarely theoretical papers can develop a full and closed theory at once, and we hope that the reviewer will take this into consideration.

---

### Official Review · Reviewer_ajzH · 2022-10-26

**Confidence:** 3
**Correctness:** 3
**Technical Novelty And Significance:** 3
**Empirical Novelty And Significance:** 3
**Recommendation:** 6

**Clarity, Quality, Novelty And Reproducibility:**

The paper is for the most part written quite clearly.  It requires no small effort to make work on the structure of neural network optimization accessible to a wider ML audience.  Bravo :)

On the novelty of their work, though Zhou et al (**Are All Losses Created Equal: A Neural Collapse Perspective**, to appear in NeurIPS 2022) also examine Neural Collapse through the lens of several losses including label smoothing, this is close enough to co-publishing that I'm willing to give the authors credit here.  In addition, this work takes a different basis for interrogating the effect on how label noise that perturbs NC structure affects generalization, which to my mind is a new (and worthy) line of inquiry.

I have an extended note about the authors' definition of memorization (defined as equation 3 in section 4.1). This concept of memorization is not unrelated to the concept by Feldman and Zhang.  Here, you are creating elements of the long tail of an anti-causal representation by adding label noise.  Your approach seems equivalent to drawing a label, then drawing observable features for this label from among the distributions of the data from differing labels, thus creating extremely improbable observations in the tail of the conditional distribution of observable features given the label.

I would ask the authors to reconsider how they define the corrupted dataset, and ask themselves if this measure defined in (3) really measures memorization.   I can see an argument where in fact a low value for ` mem` would mean that the network has memorized that the label corrupted instances should be mapped to $\mathbf{h}_{n}^{*}$, since this is the only way to achieve low training error.

A third consideration is that if the authors desire is to measure the effect of label corruption on the displacement of the data point from the class mean NC structure, it might be more informative to measure the relative distances between $\mathbf{h}_{n}^{*}$ and $\mathbf{h}_{orig}^{*}$, the mean of the true label instance.



**Strength And Weaknesses:**

### Strengths
- The authors are attempting to make the study of neural collapse applicable to more real-world models, by refining the layer-peeling model to align with the properties of many commonly used DNNs (e.g non-negativity of features).  This is important work needed to understand the training dynamics of modern networks.

- Though this paper unavoidably introduces a lot of notation, the authors do a good job in guiding readers thorugh its deployment in service of their arguments.

- The authors also explain the properties of NC very clearly, and do a good job of succinctly summarizing the development of tools for the study of neural network optimization.  Though I would add I think the study on memorization by Feldman and Zhang (NeurIPS 2020) merits mentioning here, as they tackle long tailed natural data distributions (which combined with the rigidity implied by NC means that memorization would be necessary to achieve low training error).

### Weaknesses

- In section 1, the given definition for **NC2** has a typo.  I believe it should be that the inner product between any *different* orthonormal class means (centered at $\mathbf{h}$) approaches $- \frac{1}{N - 1}$.  As written, it suggests that $<x,x> \rightarrow - \frac{1}{N - 1}$

- In section 3 just after the definition of $\mathcal{P}_{\alpha}$, it looks as if $\mathbf{y}_{n}^{(\alpha)} is defined by label smoothing, but it might be good to remind readers here, as there is a lot of notation introduced in this section.

- Could the authors clarify element (iiii) of definition 3.1?  Definition 3.1 by inspection seems to agree with the original definition of NC properties,  with the exception of (iii).  Perhaps to aid the reader here, each component of Definition 3.1 could be annotated with the NC property that it entails?  E.g  component (i) seems to entail NC1

- The first sentence in the last paragraph of section 3 is awkward (and contains a repitition "in in"). In the sentence, do the authors mean that def’n 3.1 (that includes condition (iii)) allows for Theorem 3.1 in Trier & Bruna 2022 to be established?   Could the authors please clarify.  I read this as suggesting "Definition 3.1 and the orthogonal frame result in Theorem 3.1 in …"

- The definition of the dilation quantity $\mathcal{NC}_1$ could use more motivation.  Why are we examining the trace of this product?  What intuition does this give us about NC test data? Zhu et al (which the authors cite as providing the definition) explain that $\mathcal{NC}_1$ is intended to measure the within-class variability collapse.  I presume that this is because the trace of $\Sigma_W$ will vanish as NC takes hold, while $\Sigma_B$ will approach a constant.  Perhaps a footnote could help guide the reader in the absence of Zhu et al’s explanation?

**Summary Of The Paper:**

This paper studies the phenomenon of neural collapse (NC), where representations of multi-class examples tend to collapse to the mean representation in a structured fashion.  The authors argue that the commonly used layer peeling model for NC is overly simplistic; in particular the assumption of "infinite expressivity" does not hold well with networks in practice.  They argue instead for a refinement of the layer peeling model that takes the signs of the features into account, and limits the expressiveness of the model to represent transformed inputs.  They use this model to study the interplay of memorization (which they define as the deviation from the expected NC structure of data with injected label noise) and dilation (which they define as deviation of the collapsed within-class mean structure that defines neural collapse) on the generalization ability of networks trained on both cross entropy and label smoothing.


**Summary Of The Review:**

The authors present a model that extends the Layer-Peeled model, introducing positivity conditions on the feature representations.  They develop the notions of memorization and dilation, which they use to explain the improved generalization of the label smoothing loss as compared to standard cross-entropy.

While i have some reservations about the wording of the definitions, and a more philosophical difference of opinion about memorization, I do not think either of these are serious impediments to the merit of the paper.

I will say that I hope the authors do not stop at studying the effect of label noise on NC structures. If we are to consider the penalty to generalization that NC imposes on models that train until NC onset, then surely more common issues in generalization (e.g distribution shifts) should be considered as the centre of future work.

---

> ### Author Response · Authors · 2022-11-18
> **Response to reviewer ajzH**
>
> We are very grateful to reviewer ajzH for the time taken to carefully assess our work and for the valuable feedback. Below, we address each point individually. “R” quotes the Reviewer and “A” marks the response by the authors
>
> > **R**: *"In section 1, the given definition for NC2 has a typo."*
>
> **A**: We agree with this point and fixed the typo in the revised version (see Page 2).
>
> > **R**: *"In section 3 just after the definition of $\mathcal{P}{\alpha}$ it looks as if $\mathbf{y}{n}^{(\alpha)}$ is defined by label smoothing, but it might be good to remind readers here, as there is a lot of notation introduced in this section."*
>
> **A**: We have changed the corresponding text (in page 3) as suggested. We wrote: “More precisely, given a value of alpha, the LS technique then defines the label assigned to class n as the following probability vector”.
>
> > **R**: *"Could the authors clarify element (iiii) of definition 3.1? Definition 3.1 by inspection seems to agree with the original definition of NC properties, with the exception of (iii). Perhaps to aid the reader here, each component of Definition 3.1 could be annotated with the NC property that it entails? E.g component (i) seems to entail NC1"*
>
> **A**: The NC configuration defined in Definition 3.1 indeed describes the limit of the original NC properties. We added a short subsection into the appendix to clarify this, please see Appendix B.1. on page 18-19. To sum up the discussion there, we have clarified that the component (i) entails NC1 while (ii) and (iii) together entail NC2-NC3. The component (iii) might look to be an exception as inspected by the reviewer, but together with (ii), it means that $w_n$ is proportional to $h_n-h$. Our updated subsection also discusses how the configuration in Def. 3.1 differs from the usual configuration that appeared in other papers while both configurations agree with the NC properties.
>
> > **R**: *"The first sentence in the last paragraph of section 3 is awkward (and contains a repitition "in in"). In the sentence, do the authors mean that def’n 3.1 (that includes condition (iii)) allows for Theorem 3.1 in Trier & Bruna 2022 to be established? Could the authors please clarify. I read this as suggesting "Definition 3.1 and the orthogonal frame result in Theorem 3.1 in …""*
>
> **A**: We agree that the paragraph mentioned by the reviewer is a bit confusing, hence we fixed it in the revised version (see page 4, second paragraph after Theorem 3.1). In short, with this paragraph we compared the two configurations appearing in the two results (and did not say that any of the results may imply the other). They share many similarities, especially in the structure of the feature vectors, but also have certain differences, which is due to the positivity of the features required in our model. Finally, the difference in the two results comes from the difference in the loss functions.
>
> > **R**: *"The definition of the dilation quantity NC1 could use more motivation. Why are we examining the trace of this product? What intuition does this give us about NC test data? Zhu et al (which the authors cite as providing the definition) explain that NC1  is intended to measure the within-class variability collapse. I presume that this is because the trace of ΣW  will vanish as NC takes hold, while ΣB will approach a constant. Perhaps a footnote could help guide the reader in the absence of Zhu et al’s explanation?"*
>
> **A**: Since the main paper has a page limit, we had to be very brief there and could not add a footnote as suggested by the reviewer. However, we have modified the introduction of the metric NC1 (on page 5)  according to the reviewer’s suggestion. We wrote: “This metric depicts the relative magnitude of the within-class covariance ΣW with respect to the between-class covariance ΣB of the penultimate layer features and is defined as …”.
> To provide more intuition: we agree with the reviewer’s point of view. The metric NC1 is usually used to measure variability collapse because it does not measure the absolute within-class covariance but the relative one w.r.t. the between-class covariance. This is to avoid the case that the class features seem to be clustered together, but it’s just because the whole system of all classes is compact.
>
> Also, we thank the reviewer for the valuable notes. We think that the notes are very helpful and will take them into consideration in our future research on this topic. Concerning the second note, we would mention that h_n^k is the feature vector corresponding to an example originally from class n with a corrupted label as another class, say p, and h_n^* is the mean of test features of class n. The network attempts to memorize the sample of h_n^k as class p, and hence pushes the feature vector far away from its original class n. Hence h_n^k should not be mapped to h_n^*, but to the class p. We measure the memorization by how far away the feature representation is pushed from its original class.

---

> > ### Comment · Reviewer_ajzH · 2022-11-22
> > **Response to authors**
> >
> > Thank you for taking my comments into account, and for clarifying some of my questions.
> >
> > > Concerning the second note, we would mention that h_n^k is the feature vector corresponding to an example originally from class n with a corrupted label as another class, say p, and h_n^* is the mean of test features of class n. The network attempts to memorize the sample of h_n^k as class p, and hence pushes the feature vector far away from its original class n. Hence h_n^k should not be mapped to h_n^*, but to the class p. We measure the memorization by how far away the feature representation is pushed from its original class.
> >
> > Yes, I believe I understand what you mean by your description of memorization.  However, I remain unconvinced that simply measuring the distance from the original class is sufficiently informative.  Measuring the ratio of the distance of the label-perturbed data point from the original class to the perturbed class (or perhaps projecting the datapoint onto the vector between these classes) would be a more appropriate statistic.

---

> > > ### Author Response · Authors · 2022-11-24
> > > **Response to reviewer**
> > >
> > > Thank you for clarifying your idea. In our experiments, we actually tried different metrics for both memorization and dilation, including different normalization definitions, and only the metrics reported in our paper lead to a trend that is very close to linear, and the memorization-dilation curves of CE and LS coincide. We hence realized that there is something special about our particular definitions of memorization and dilation. Conversely, defining memorization via the suggested normalized definition unfortunately does not lead to such a trend that coincides for CE and LS. Since we aimed for a theoretical model that is as close as possible to the practical observations, we chose the reported metrics, which we think are also still very interpretable, as we explained in our paper.

---

### Official Review · Reviewer_ww8W · 2022-10-26

**Confidence:** 5
**Correctness:** 2
**Technical Novelty And Significance:** 3
**Empirical Novelty And Significance:** 3
**Recommendation:** 6

**Clarity, Quality, Novelty And Reproducibility:**

The study focuses on an important problem. The result is somewhat inspiring and interesting. The clarity of this paper needs to be improved, especially for the structure.

**Strength And Weaknesses:**

Strengths:

The study gives the solution structure of the LS loss, whereas previous studies mainly focus on the original CE loss.
The study gives an explanation of the better generalization of LS loss than CE loss under label corruption.


Weaknesses:

- Misleading description. The authors claim to address the model with noisy data. However, they actually only consider label corruption. Noisy data does not equivalently refer to label noise. A more precise description should be adopted.

- The positivity constraint is confusing. Although the last layer will have positive feature when ReLU is performed, a neural network does not necessarily end with a ReLU activation in most cases. If an identity connection or a BN layer is appended, we can easily get rid of the positivity constraint. More importantly, I do not see a necessary connection between the positivity constraint and the later analysis of label corruption. Is your result (LS loss shows less memorization and dilation) valid only when the positivity constraint is accompanied?

- The claim that less dilation leads to better generalization lacks rigorous support. As indicated by your definition, dilation reflects the compactness and separation of test features. But generalization ability seems to be more related to loss and accuracy on test set. So, a more rigorous relation between “dilation” and the “generalization” in your context should be constructed. Otherwise, the claim seems to be groundless.

- The memorization-dilation model is confusing. First, it only considers two classes, which is unrealistic. Why do the authors only consider two classes? I do not think it would be a simple extension by generalizing the two-class result into multiple classes. Besides, why are H and W fixed as the optimal solution in the memorization-dilation model? Its motivation and rationality are unclear and need more discussion.

- The second equation in (NC2) in page2 is wrong.

- I suggest that the authors consider more cases in Theorem 3.2. It would be better if the authors could first give the solution structure of the LS loss without positivity constraint, and the CE loss with the positivity constraint, and then deal with the LS loss with positivity constraint as stated in Theorem 3.2.



------ After rebuttal

The authors address most of my questions and concerns. I increase my score to 6.




**Summary Of The Paper:**

This study gives the solution structure of the label smoothing (LS) loss with positivity constraint, which is an orthogonal variant of the original neural collapse configurations. The paper further explores why LS loss has a better generalization than CE loss when label corruption emerges based on the memorization-dilation model. It reveals that LS loss induces less memorization, so leads to less dilation, which explains its better generalization.

**Summary Of The Review:**

The paper suffers from some issues, as listed in the weaknesses. Most importantly, the main claim lacks rigorous supports. Some settings in the model are confusing.

---

> ### Author Response · Authors · 2022-11-18
> **Response to the reviewer ww8W**
>
> We are very grateful to reviewer ww8W for the time taken to carefully assess our work and for the valuable feedback. Below, we address each point individually. “R” quotes the Reviewer and “A” marks the response by the authors:
>
> > **R**: *"Misleading description. The authors claim to address the model with noisy data. However, they actually only consider label corruption. Noisy data does not equivalently refer to label noise. A more precise description should be adopted."*
>
> **A**: We have changed the title to “Memorization-Dilation: Modeling Neural Collapse under Label Noise”.
>
> >**R**: *"The positivity constraint is confusing. Although the last layer will have positive feature when ReLU is performed, a neural network does not necessarily end with a ReLU activation in most cases. If an identity connection or a BN layer is appended, we can easily get rid of the positivity constraint. More importantly, I do not see a necessary connection between the positivity constraint and the later analysis of label corruption. Is your result (LS loss shows less memorization and dilation) valid only when the positivity constraint is accompanied?"*
>
> **A**: To be clear, we do not end the network with ReLU, there is one last ReLU operation in the part of the network that we call the feature embedding, and then we apply another linear layer that we interpret as (or call) the classifier. Adding any additional linear layers without ReLU would be an overparameterization of just one single linear layer.
> Concerning the connection between the positivity constraint and the memorization-dilation model, the result can be shown to still be valid in absence of positivity constraint (however the NC configuration becomes different). The proof for the memorization-dilation model will then be simpler in comparison to the case of positive features. Nevertheless, we focus on the case of positive features, which we believe represent real-life networks better than the case of general features. Moreover, if we developed the theory for both cases, the paper would be too long.
>
> > **R**: *"The claim that less dilation leads to better generalization lacks rigorous support. As indicated by your definition, dilation reflects the compactness and separation of test features. But generalization ability seems to be more related to loss and accuracy on test set. So, a more rigorous relation between “dilation” and the “generalization” in your context should be constructed. Otherwise, the claim seems to be groundless."*
>
> **A**: The relation between dilation and generalization was explained in Section 4.2 (the second paragraph, Page 7 top). In the revised version, we however added a citation, namely Kornblith et al. (2021), to the paragraph as a support for our claim that less dilation leads to better class separation and hence accounts for improvement in accuracy. This statement has been justified in the cited paper.
>
> > **R**: *"The memorization-dilation model is confusing. First, it only considers two classes, which is unrealistic. Why do the authors only consider two classes? I do not think it would be a simple extension by generalizing the two-class result into multiple classes. Besides, why are H and W fixed as the optimal solution in the memorization-dilation model? Its motivation and rationality are unclear and need more discussion."*
>
> **A**: In general, we aimed for a theoretical work with rigorous proofs, and hence we had to restrict ourselves to a simplified setting. Indeed, we do not expect and did not claim that the extension to the multi-class setting is trivial. We instead expect our work to be the first step in a sequence of papers, by ourselves and others, that will build upon each other and make the learning scenario more and more realistic. The study of binary classification problems is the starting point, based on which one could further explore the more general cases, including multi-class setting. In fact, the experimental results for the multi-class case look promising (see Figure 4,5,6 in the appendix), however we could not explain them rigorously like the binary case and had to leave this for future research. Also, we assumed for simplicity that H and W are at neural collapse configurations, which we think is realistic as is observed by the experiments. We have added a sentence with this explanation to the revised paper (the last sentence in the first paragraph below Def. 4.1, page 7).
> We were not claiming that we developed a complete and close theory, only rare theoretical papers can do so at once. However the analysis has to start somewhere, and we hope that future works will take it further, making the model more and more realistic.

---

> > ### Author Response · Authors · 2022-11-18
> > **Response to reviewer ww8W (2/2)**
> >
> > > **R**: *"The second equation in (NC2) in page2 is wrong."*
> >
> > **A**: We agree that in this equation the indices m and n were exchanged at several positions. We fixed the typo in the revised version, page 2.
> >
> > >**R**: *"I suggest that the authors consider more cases in Theorem 3.2. It would be better if the authors could first give the solution structure of the LS loss without positivity constraint, and the CE loss with the positivity constraint, and then deal with the LS loss with positivity constraint as stated in Theorem 3.2."*
> >
> > **A**: We find the reviewer’s suggestion helpful and have added a subsection in the appendix (namely B.1, page 18-19) introducing the usual version (without positivity) and discussing its difference to our positivity version. There are however several reasons for us to put both the LS loss and positivity constraint into a compact theorem in the main paper. First, the CE loss can be seen as a special case of LS, where the LS parameter is set to 0. The proof for the CE loss might be a bit simpler but not significantly, and can be also seen as a special case of the LS loss case. Furthermore, as we also mentioned in our paper, the problem without positivity has been studied by several other papers (first for CE, and in a concurrent work also for LS), and hence it is not necessary to restate proven and well-known results. Most importantly, since the main paper has a page limit, we have to merge those cases into a general compact version that is short and does not require too much space.
> >
> >
> > Concerning the correctness of the paper, we hope that we convinced the reviewer that the true errors were only typos, and are now fixed.

---

> ### Comment · Reviewer_ww8W · 2022-12-07
> **thanks for the response**
>
> Thanks for the detailed response. To me, the 2nd and the 4th concerns are still not resolved.
>
> I think the authors did not answer my 2nd question in a straight way. The authors claim that the proof without positivity constraint will be simpler. But why do they not deal with the more general case without the positivity constraint? I do not think positive features more represent the real-life networks. As I said, most networks do not end with positive features. So, the motivation of positivity constraint, and its connection to the contribution, are still unclear.
>
> The work of two-class case is a contribution. But the authors also agree that the extension to multi-class setting is not a trivial problem. So, this limitation impedes the significance of this study. Besides, it is still confusing why H and W are fixed as the optimal solution in the memorization-dilation model. The authors only respond that it is realistic as observed by experiments. Where are the experiment results and what is the indication? Its motivation and rationality are still unclear and need more discussion. I think the authors should offer a straight explanation, instead of arguing that the work cannot be complete and the theory cannot be close at once.
>
> Besides, I notice that a lot of neural collapse studies are not cited and discussed. I hope the authors could fix this problem in the revised paper.
>
> I would like to keep my original rating for now, but I do not object if this paper is accepted.

---

> ### Author Response · Authors · 2022-12-12
> **Response to reviewer**
>
> Thanks for the response. We would like to partially repeat our response from the last time, yet hopefully more clearly. Concerning the 2nd question, the positive features model is itself a contribution. We do not agree that most networks do not end with positive features. In converse, we think that many common practical networks do not apply techniques like batch normalization. For example, a vanilla fully connected ReLU network is an important object of study and not an esoteric thing. For such a network, the penultimate layer features are indeed positive and hence it is important to study this case.
>
> In fact, the positivity constraint on the features has been also discussed in [Zhu et al. 2021] (page 10), where the authors show briefly how to compensate for the positivity using the bias vector. Note however that a rigorous result for the positive case has not been stated, as the resulting configuration is not a minimizer to the corresponding model. Taking a different approach, [Tirer & Bruna 2022] also incorporates the features positivity into their model by directly adding the ReLU nonlinearity.
>
> Concerning the 4th question, there are many theoretical works on binary classification and this is considered an important and worthwhile topic of study. Also, fixing $W$ and $H$ is a simplifying assumption in the sense that it helps us to complete the proof. One may certainly drop this assumption in the model and consider $W$ and $H$ as minimization variables, but this would lead to a much more difficult problem. A reasonable point to fix these variables is their optimal values when the dilation becomes zero. In other words, we study an alternative of the minimization problem over the variables $W$, $H$ and $r$ where we first optimize over $W,H$ and then turn to $r$. We know that this will be more or less different from the original problem, but as we said some assumptions must be made for the first steps of analysis and we hope that the next steps will make it more realistic. Also, in our experiments we saw that $W$ and $H$ approximately form the dual simplex equiangular tight frame configuration proved in Theorem 3.2. The corresponding experimental result for the binary case has been shown in Figure 4 in the appendix.

---

### Official Review · Reviewer_7FKE · 2022-10-30

**Confidence:** 4
**Correctness:** 4
**Technical Novelty And Significance:** 3
**Empirical Novelty And Significance:** 3
**Recommendation:** 6

**Clarity, Quality, Novelty And Reproducibility:**

Overall, the paper is well-organized and well-written. The presentation of section 4.2, particularly Denifition 4.1, could be improved. For example, $u_1$ and $u_2$ could be introduced right before Definition 4.1. The results on nonnegative features extend previous work on MSE to label smoothing. The memorization-dilation model is new and could provide further insight into the connection between neural collapse and generalization.

**Strength And Weaknesses:**

## Strength:
- This paper formally shows NC solutions for label smoothing and cross-entropy (CE) under nonnegative features. Nonnegative features are only considered for MSE loss in the previous work.
- An interesting label-corrupted experiment is proposed, and the linear relation between dilation and memorization when corrupted levels are not large is very interesting.
- The proposed memorization-dilation model may provide further insight into the connection between neural collapse and generalization.

## Weakness:
- As the nonnegative features model has already been studied in Tirer & Bruna (2022), though for the MSE loss, the authors may highlight the technical challenges in extending the results to label smoothing.
- This paper shows that on training data with label noise, label smoothing could be more robust to label noise and achieve better performance on the original testing data. Based on this, the paper claims in many places that label smoothing leads to improved generalization in classification tasks. How does this better performance under label noise translate to better generalization in the standard case without label noise?
- Following the above point, the recent work [A] shows that label smoothing and CE indeed produce similar performance when the network is sufficiently large and trained sufficiently long in the standard way without label noise. Since label smoothing and CE produce similar NC features under the unconstrained feature models, is the better performance of label smoothing on label noise because it has a different convergence speed than CE? It will be of interest to perform experiments with more iterations and see whether the results are the same.
- It is interesting to note that MSE has better testing performance in Figure 1. Could the authors provide some comments on this?
- Does the memorization-dilation model only consider one mislabeled training sample per class?
- Could the memorization-dilation model be extended to the multi-class case?

[A] Zhou et al., Are All Losses Created Equal: A Neural Collapse Perspective; arXiv preprint arXiv:2210.02192; 2022.



**Summary Of The Paper:**

The authors study the phenomenon of neural collapse (NC) under several variants of the layer-peeled model.  Since features from modern networks are the outcome of some non-negative activation functions, such as ReLU, the paper first considers the case of non-negative features and shows that label smoothing also produces NC solutions in this case. The authors then propose a new memorization-dilation model on test data with randomly label-corrupted training samples, which shows a linear relation between NC degree (dilation) and overall distance between test samples and their corresponding class-mean (memorization).  Finally, in the label-corrupted setting, they formally prove the advantage of label smoothing over cross-entropy in binary classification tasks, and show some supporting experiments.

**Summary Of The Review:**

This paper is well-organized and provides interesting theoretical and empirical results on neural collapse with label smoothing. The memorization-dilation model could provide further insight into the connection between neural collapse and generalization, though it is not clear how the performance with label noise can be used to understand the generalization performance in the standard setting without label noise. I look forward to the authors’ response to revise my decision.

---

> ### Author Response · Authors · 2022-11-18
> **Response to reviewer 7FKE (1/2)**
>
> We are very grateful to reviewer 7FKE for the time taken to carefully assess our work and for the valuable feedback. Below, we address each point individually. “R” quotes the Reviewer and “A” marks the response by the authors:
> > **R**: *"As the nonnegative features model has already been studied in Tirer & Bruna (2022), though for the MSE loss, the authors may highlight the technical challenges in extending the results to label smoothing."*
>
> **A**: From the technical perspective, we do not see our result as an extension from the nonnegative features model in Tirer & Bruna (2022) with MSE loss to label smoothing. Their model is studied as a version of the so-called extended unconstrained features model, where one more layer is added to the layer-peeled model. In presence of ReLU, the corresponding result strongly requires an equality between the nuclear norm of the feature vector before and after applying ReLU. In our paper, we took a different approach by not extending the model for one layer, but directly requiring the nonnegativity to the features.
>
> On the other hand, we think that our proof technique is more related to the one in [A] (which is also mentioned by the reviewer), and also the one in Zhu et al. (2021). In this regard, the main difficulty is how to deal with the nonnegativity constraints, as these do not allow us to get a simple first-order optimality condition like in the mentioned works.
>
> To make our contribution clear, and since the main paper has a page limit, we decided to add a short discussion about this to the appendix section that contains the corresponding proof (see page 17-18 in the appendix).
>
> > **R**: *"This paper shows that on training data with label noise, label smoothing could be more robust to label noise and achieve better performance on the original testing data. Based on this, the paper claims in many places that label smoothing leads to improved generalization in classification tasks. How does this better performance under label noise translate to better generalization in the standard case without label noise?"*
>
> **A**: We added a paragraph explaining this issue (see page 9, just after the theorem):
>
> “Note that in practice, the data often have noise in the sense that not all examples are perfectly labeled. More importantly, examples from different classes may share many similarities, a situation that is also covered by the MD model: the feature representations of samples from those classes are biased toward each other. In this case, LS also leads to decreased dilation which corresponds to better class separation and higher performance Kornblith et al. (2021)”

---

> > ### Author Response · Authors · 2022-11-18
> > **Response to reviewer 7FKE (2/2)**
> >
> > > **R**: *"Following the above point, the recent work [A] shows that label smoothing and CE indeed produce similar performance when the network is sufficiently large and trained sufficiently long in the standard way without label noise. Since label smoothing and CE produce similar NC features under the unconstrained feature models, is the better performance of label smoothing on label noise because it has a different convergence speed than CE? It will be of interest to perform experiments with more iterations and see whether the results are the same."*
> >
> > **A**: Our experiments were performed until convergence. We added a short sentence clarifying this to the revised version (more precisely to the paragraph introducing the experimental setting in page 5).
> >
> > Concerning the reviewer’s question, since we performed the training until convergence, the better performance of LS over CE in the experiments is not due to the convergence speed. As pointed out in our theoretical study via Theorem 4.3, in the presence of noise LS would lead to lower dilation and hence achieve better generalization in comparison to CE. Note that to realize the limited expressivity in the theory (as mentioned in the first paragraph of Section 4.2), we performed experiments using a moderate network architecture (see our experimental setup in page 5), and not a very large one as in [A], which might have approximately infinite expressivity and can almost fit perfectly the NC configuration using both losses, i.e. achieve almost zero dilation.
> >
> > We also added a paragraph briefly discussing the difference between our result and the claim in [A] (in the revised paper as Zhou et al. (2022b)), namely the second paragraph after Theorem 4.3 (page 9):
> >
> > “Interestingly, the concurrent work Zhou et al. (2022b) has shown that in the noiseless setting CE andLS lead to largely identical test accuracy, which seems to contradict the statement that LS performs better claimed by our work as well as many others, e.g. Kornblith et al. (2021); Müller et al. (2019). However, note that Zhou et al. (2022b) requires the network to be sufficiently large so that it has enough expressive power to fit the underlying mapping from input data to targets, as well as to be trained until convergence. While the latter is easy to obtain, it is difficult even to check if the first requirement holds. The difference between the two results is hence possibly caused by the effect of noise and by the network expressivity: while we aim to model the limited expressivity by the MD relation, Zhou et al. (2022b) focuses on networks with approximately infinite expressivity.”
> >
> > > **R**: *"It is interesting to note that MSE has better testing performance in Figure 1. Could the authors provide some comments on this?"*
> >
> > **A**: In our paper, we only focus on CE and LS loss, while the performance of MSE is just a side result. It is in fact interesting to observe that in the noisy case MSE performs better than other losses, but we could not yet prove this phenomenon using our MD model. As said in the conclusion, we do believe that the MD can be proven also for MSE, but coming up with such a proof is left for future work.
> >
> > > **R**: *"Does the memorization-dilation model only consider one mislabeled training sample per class?"*
> >
> > **A**: No, there are many mislabeled training samples in each class. Remember that the model has a parameter $\eta$ controlling the noise level, while the point $u_i$ for each $i=1,2$ refers to the representative of a distribution of the class features instead of being an only single feature representation. Please see also the footnote in page 7 of the revised paper (or page 8 of the original paper).
> >
> > > **R** : *"Could the memorization-dilation model be extended to the multi-class case?"*
> >
> > **A**: As said in the conclusion, our MD model is limited to the binary classification case. Since the experimental results look promising (see e.g. Figures 4,5,6 in the appendix, page 14-15), we hope to prove an extension to the multi-class case in future work, but such an extension is not trivial.

---

> ### Author Response · Authors · 2022-12-12
> **Reminder**
>
> Thank you again very much for your careful reading of our manuscript. Since the Stage 2 deadline is approaching, this is a gentle reminder to acknowledge our revision and response, in which we hopefully gave answers to all of your questions and suggestions.

---

### Decision · Program_Chairs · 2023-01-20

**Decision:**

Accept: poster

**Justification For Why Not Higher Score:**

Analysis is limited to two-class classification and the paper doesn't explore alternative definitions of memorizations adequately

**Justification For Why Not Lower Score:**

Contributions to understanding label smoothing is interesting.

**Metareview: Summary, Strengths And Weaknesses:**

This paper extends the layer-peeled model for studying neural collapse to a more realistic version, studies the relationship between memorization and neural collapse which finally leads to a better understanding of label smoothing. The main limitation of the work is that the analysis is limited to 2-classes. However, the contributions of the paper are significant enough to be accepted for publication.

**Note From Pc:**

if the above contains the word "oral" or "spotlight" please see: "oral" presentation means -> notable-top-5% and "spotlight" means -> notable-top-25%. As stated in our emails, we are disassociating presentation type from AC recommendations

**Summary Of Ac-Reviewer Meeting:**

The main contributions of the paper, its strengths and weaknesses were discussed.

ajzH: Pro: Contrary to many papers on generalization, this paper was very accessible. Con: I wasn’t totally convinced the definition of memorization. Maybe it shouldn't be called memorization

7FKE: Pro: The connection between memorization and neural collapse is interesting. I also liked the contribution about non-negative features. Con: memorization of two classes is very limited.

ww8W: ​​I have responded to the authors in the openreview. I keep my score unchanged because my 2nd and 4th concerns are still NOT resolved. See my reply in the openreview for details. I may make more discussion based on the authors' following reply. I do not object if this paper is accepted.